# Prognosis Following Sustained Virologic Response in Korean Chronic Hepatitis C Patients Treated with Sofosbuvir-Based Treatment: Data from a Multicenter Prospective Observational Study up to 7 Years

**DOI:** 10.3390/medicina60071132

**Published:** 2024-07-14

**Authors:** Yewan Park, Seong-Kyun Na, Jae-Hyun Yoon, Sung-Eun Kim, Ji-Won Park, Gi-Ae Kim, Hyo-Young Lee, Young-Sun Lee, Jeong-Han Kim

**Affiliations:** 1Department of Internal Medicine, Kyung Hee University School of Medicine, Seoul 02447, Republic of Korea; yewanish@gmail.com (Y.P.); antiankle@hanmail.net (G.-A.K.); 2Department of Internal Medicine, Inje University Sanggye Paik Hospital, Seoul 01757, Republic of Korea; drcoramdeo@naver.com (S.-K.N.); catchhyong@gmail.com (H.-Y.L.); 3Department of Internal Medicine, School of Medicine, Chonnam National University Hospital, Gwangju 61469, Republic of Korea; zenmake14@gmail.com; 4Department of Internal Medicine, Hallym University College of Medicine, Anyang 14068, Republic of Korea; sekim@hallym.or.kr (S.-E.K.); miunorijw@hallym.or.kr (J.-W.P.); 5Department of Internal Medicine, Korea University Medical Center, Seoul 08308, Republic of Korea; lys810@korea.ac.kr; 6Department of Internal Medicine, Konkuk University School of Medicine, Seoul 05030, Republic of Korea; 7Research Institute of Medical Science, Konkuk University School of Medicine, Seoul 05029, Republic of Korea

**Keywords:** hepatitis C, direct-acting antiviral, sustained virologic response, hepatocellular carcinoma, sofosbuvir, ledipasvir

## Abstract

*Background and Objectives*: Chronic hepatitis C (CHC) can be cured with direct-acting antiviral (DAA) therapy. In Korea, sofosbuvir (SOF) and ledipasvir (LDV)/SOF were launched in 2016. Patients who achieve a sustained virologic response (SVR) following DAA treatment are predicted to have a favorable prognosis. Nevertheless, little is known regarding the prognosis of Korean CHC patients who receive SOF-based treatment and achieve SVR. Therefore, the purpose of this study was to look into the long-term outcomes for these patients. *Materials and Methods*: This was a prospective, multicenter observational study. CHC patients were enrolled who, following SOF or LDV/SOF treatment, had achieved SVR. The last day for follow-up was December 2023. The primary endpoint was HCC occurrence, which was checked at least once per year. *Results*: A total of 516 patients were included in this analysis, with a median follow-up duration of 39.0 months. Among them, 231 were male patients (44.8%), with a median age of 62.0 years. Genotypes were 1 (90, 17.4%), 2 (423, 82.0%), and 3 (3, 0.6%). The combination of SOF plus ribavirin was the most common treatment (394, 76.4%). In total, 160 patients were cirrhotic (31.0%), and the mean Child–Pugh score was 5.1. Within a maximum of 7 years, 21 patients (4.1%) developed HCC. Patients with HCC were older (69 vs. 61 years, *p* = 0.013) and had a higher cirrhosis incidence (81.0 vs. 28.9%, *p* < 0.001), higher AFP (6.0 vs. 3.3, *p* = 0.003) and higher APRI (0.8 vs. 0.5, *p* = 0.005). Age over 65 (*p* = 0.016) and cirrhosis (*p* = 0.005) were found to be significant risk factors for HCC by Cox regression analysis. *Conclusions*: Patients who achieved SVR with SOF-based treatment had a relatively favorable prognosis. However, the risk of HCC was not eliminated, especially in older and cirrhotic patients. Therefore, routine follow-up, surveillance, and early treatment are required.

## 1. Introduction

Chronic hepatitis C (CHC) remains a significant public health problem, with an estimated 50 million people currently dealing with infection [1,2,3]. Untreated CHC progresses to cirrhosis; the incidence of HCC in patients with cirrhosis is reported to be 1–5% per year, and the incidence of decompensation is 3–6% per year [4]. With effective direct-acting antivirals (DAAs), a sustained virologic response (SVR) has been achieved in more than 90% of cases. However, the risk of HCC is not completely reduced even after SVR [5,6].

Since sofosbuvir (SOF) received food and drug administration (FDA) approval in 2013, it has been used as the primary treatment for CHC. Several studies have reported good clinical outcomes with SOF-based therapy. However, most of these studies were conducted in Western countries, and long-term outcomes other than those of HCC have not been fully described [5,6,7,8].

Before the launch of pan-genotypic DAAs, SOF/ribavirin (RBV) or ledipasvir (LDV)/SOF treatment was widely used in Korea. Several studies have reported the effectiveness of SOF-based therapy [9,10]. However, studies analyzing HCC occurrence and long-term outcomes are rare.

Therefore, the purpose of this study was to look into the long-term outcomes for patients with CHC who had SOF-based treatment and obtained SVR, including HCC occurrence, changes in liver function, development of hepatic decompensation, and reinfection or recurrence of HCV.

## 2. Materials and Methods

### 2.1. Patients

Six South Korean university hospitals participated in this prospective observational multicenter study, which was authorized by the ethical boards of each facility. All participants provided informed consent. CHC patients who achieved an SVR after SOF-based treatment were enrolled between August 2016 and February 2021. Patients were followed-up until December 2023. The following were the inclusion criteria: (1) CHC patients with the achievement of SVR after SOF or LDV/SOF or daclatasvir (DCV)/SOF regimen, and (2) age ≥ 18 years. The following were the exclusion criteria: (1) patients who declined enrollment; (2) a lack of comprehension or capacity to provide informed consent; (3) the existence of malignant tumors, such as HCC, that had already been diagnosed; (4) concurrent infection with human immunodeficiency virus or chronic hepatitis B; (5) patients who underwent liver transplantation; and (6) a history of previous CHC treatment.

SVR was defined as HCV RNA levels that were undetectable 12 weeks following the end of treatment. Liver pathology, radiological findings (liver configuration, splenomegaly, ascites, and esophageal or gastric varices), or clinical findings (symptoms and signs of cirrhosis or its complications) were used to define cirrhosis.

The degree of hepatic fibrosis was assessed using the fibrosis-4 index (FIB-4) and the aspartate aminotransferase (AST) to platelet ratio index (APRI). The formulas listed below were used to determine these markers: APRI = ([AST level/AST upper limit of normal]/platelet count [10^9^/L]) × 100; FIB-4 index = age (years) × AST (IU/L)/(platelet count (10^9^/L) × (alanine aminotransferase [IU/L]^1/2^). Baseline and yearly evaluations of the APRI and FIB-4 scores were conducted.

The primary endpoint was HCC development, and the secondary endpoints were HCV reinfection or recurrence and the development of decompensation. Overt ascites, hepatic encephalopathy, variceal bleeding, and impairment of liver function with serum bilirubin levels ≥ 3 mg/dL were considered indicators of liver decompensation. All enrolled patients had liver function tests, tests for HCV RNA and alpha-fetoprotein (AFP) levels, and radiological exams such as computed tomography or abdominal ultrasonography at least once a year during the follow-up period.

### 2.2. Statistical Analysis

We used descriptive statistics to describe the baseline demographics of the patients. Because the data are not normally distributed, continuous variables are expressed as medians (interquartile ranges), while categorical variables are presented as numbers (%). To compare variables between groups, Mann–Whitney U, chi-square, and Fisher’s exact tests were conducted. Significant changes in liver function and noninvasive fibrosis scores were evaluated using the Wilcoxon signed-rank test at two distinct time points. The Kaplan–Meier method was used to assess the cumulative incidence of HCC, and the log-rank test was used to identify any differences.

Cox regression hazard model (forward) univariate and multivariate analyses were used to find independent predictors of HCC. All analyses were carried out using MedCalc (version 22.005; MedCalc Software, Mariakerke, Belgium) and SPSS (version 27.0; IBM Corp., Armonk, NY, USA), with statistical significance set at *p* < 0.05.

## 3. Results

We analyzed 516 Korean patients with CHC who achieved sustained SVR after SOF or LDV/SOF treatment. The cohort included 44.8% males, with a median age of 62 years (Table 1). The median follow-up duration was 39.0 months. Twenty-one patients (4.1%) developed HCC post-SVR. Compared to those without HCC, these patients were older (>65 years), more likely to have cirrhosis, and exhibited higher AFP levels and liver fibrosis scores.

One patient experienced HCV relapse or reinfection 31 months after SVR, demonstrating the importance of continued surveillance even after achieving SVR (Appendix A). Decompensation events occurred in a small but significant subset of the cohort, correlating with higher baseline scores of MELD, APRI, and FIB-4, particularly among patients with cirrhosis or those of older age (Appendix A).

Liver function and fibrosis markers showed notable trends over time. Child–Pugh scores remained stable, indicating consistent liver function across the cohort. Conversely, the MELD scores started to decline steadily in the second year following SVR, suggesting an improvement in liver disease severity among the patients (Table 2). Similarly, both APRI and FIB-4 scores demonstrated significant reductions compared to the pre-treatment scores, underlining the therapeutic benefit of SOF-based treatment in reducing liver fibrosis (Table 3).

HCC occurred in 21 patients (Table 4). Risk factor analysis for HCC occurrence highlighted age > 65 years and the presence of cirrhosis as significant predictors (Table 5, Figure 1). Furthermore, patients developing HCC had higher AFP levels and APRI scores compared to those who did not develop cancer.

## 4. Discussion

Our multicenter prospective observational study evaluated 516 Korean patients treated with SOF-based regimens for CHC, focusing on the long-term prognosis and HCC incidence after DAA therapy. We found that SVR was consistent with global DAA success, signaling a significant leap in HCV management. Despite the effectiveness of DAAs in viral eradication and liver function improvement, as indicated by reduced Child–Pugh and MELD scores, along with better noninvasive fibrosis scores (APRI, FIB-4), the 4.1% HCC occurrence in our cohort highlights the ongoing cancer risk after achieving SVR, notably in older patients and those with preexisting cirrhosis.

Given the effectiveness of DAAs in the global anti-HCV battle, the eradication of hepatitis C has become an attainable goal [11]. In our study, the effectiveness of these treatments was mirrored in a Korean patient cohort, aligning with global data showcasing the success of DAAs across diverse populations and clinical settings [12,13,14,15,16,17,18]. This efficacy is not only a testament to the robustness of these therapeutic agents but also highlights their critical role in the World Health Organization’s aim to eliminate HCV as a public health threat by 2030 [1]. By integrating our findings with data from previous studies, our research further expands the evidence supporting DAAs as the main drugs in the quest to eradicate HCV infection.

Throughout our study, consistent stabilization of liver function and noninvasive fibrosis markers was observed. This stability, evidenced by steady Child–Pugh and MELD scores alongside noninvasive fibrosis indicators such as APRI and FIB-4, underscores the enduring impact of DAA therapy beyond achieving SVR. The persistent normalization of these markers signifies the halt of liver disease progression and a reversal of liver fibrosis in some patients [19,20,21]. This phenomenon is particularly noteworthy, as it highlights the potential of DAAs to confer long-term liver health benefits and mitigate the risk of future liver-related complications.

Aging and liver cirrhosis have been identified as significant risk factors for the development of HCC after SVR, highlighting critical areas for post-treatment management and surveillance. The association between aging and HCC risk has been well documented and was identified as a risk factor in the era of interferon-based treatments. This was further corroborated by a recent study employing machine learning to analyze outcomes after DAA therapy [22,23]. The aging process is inherently associated with cumulative cellular damage, and older patients often have a longer history of HCV infection, leading to more advanced liver fibrosis. Despite these insights, current international guidelines lack a consensus on post-treatment HCC surveillance and fail to offer unified strategies addressing the risks associated with aging [11,24,25]. This gap highlights the urgent need for a consensus-driven, evidence-based approach to surveillance, emphasizing personalized monitoring to better manage the complex interplay of factors influencing HCC development in the post-treatment landscape. This approach is crucial for enhancing early detection, optimizing patient management, and ensuring that the benefits of DAA therapy extend beyond viral clearance for comprehensive liver health and cancer prevention.

Acknowledging the limitations of this study is essential for a comprehensive understanding and interpretation of the findings. Our research focused on a specific patient population within Korea treated with SOF-based regimens, potentially limiting the generalizability of our results to other HCV genotypes or patient populations with different demographic or clinical characteristics. Additionally, the potential for confounding factors could not be entirely excluded despite rigorous analytical methods. The observational nature of our study design necessitates a cautious interpretation of the inferred causal relationships. These limitations highlight the need for further research, including larger multicenter studies across diverse populations and with various DAA regimens, to validate and expand our findings. By acknowledging these aspects, we invite the scientific community to build on our work, aiming for a fuller understanding of the natural course of CHC after DAA therapy and the persistent challenges in the quest for HCV eradication.

## 5. Conclusions

In conclusion, our study highlighted the long-term outcomes of patients who achieved SVR following SOF-based therapy. Despite sustained improvements in liver function and fibrosis markers, the risk of HCC remains after eradication, especially in elderly patients and those with cirrhosis. Consequently, it is necessary to implement careful follow-up strategies based on each individual’s risk predisposition.

## Figures and Tables

**Figure 1 medicina-60-01132-f001:**
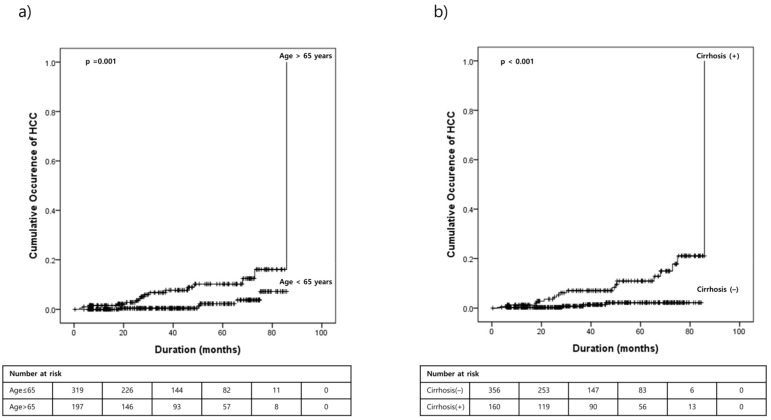
Cumulative occurrence of HCC after SVR according to (**a**) age > 65 years or not and (**b**) cirrhosis or not. HCC, hepatocellular carcinoma; SVR, sustained virologic response.

**Table 1 medicina-60-01132-t001:** Characteristic data of patients.

	Total (*n* = 516)	HCC Occurence (−) (*n* = 495)	HCC Occurence (+) (*n* = 21)	*p*-Value
Male	231 (44.8%)	219 (44.2%)	12 (57.1%)	0.173
Age (years)	62.0 (55.0–70.0)	61 (55–70)	69 (66–71)	0.013
Genotype				<0.001
1	90 (17.4%)	86 (17.4%)	4 (19.0%)	
2	423 (82.0%)	408 (82.4%)	15 (71.4%)	
3	3 (0.6%)	1 (0.2%)	2 (9.5%)	
Follow-up duration (months)	39.0 (18.6–62.3)	38.4 (18.3–61.7)	57.5 (40.6–73.8)	0.002
AFP (ng/mL)	3.4 (2.2–6.0)	3.3 (2.2–5.8)	6.0 (4.3–9.0)	0.003
Pre-treatment HCV RNA (IU/mL)	680,000 (94,050–3,005,000)	761,000 (94,225–3,042,500)	449,000 (100,000–670,000)	0.169
Cirrhosis	160 (31.0%)	143 (28.9%)	17 (81.0%)	<0.001
Significant alcohol consumption	53 (10.3%)	51 (10.3%)	2 (9.5%)	0.632
Child–Pugh score	5.0 (5.0–5.0)	5.0 (5.0–5.0)	5.0 (5.0–5.0)	0.822
5	486 (94.2%)	466 (94.1%)	20 (95.2%)	
6	22 (4.3%)	21 (4.2%)	1 (4.8%)	0.996
7	3 (0.6%)	3 (0.6%)	0 (0%)	
8	2 (0.4%)	2 (0.4%)	0 (0%)	
9	1 (0.2%)	1 (0.2%)	0 (0%)	
10	2 (0.4%)	2 (0.4%)	0 (0%)	
MELD	7.0 (6.0–8.0)	7.0 (6.0–8.0)	7.0 (7.0–9.0)	0.449
APRI	0.5 (0.3–1.1)	0.5 (0.3–1.1)	0.8 (0.7–1.5)	0.005
FIB-4	2.6 (1.7–5.0)	2.5 (1.7–4.8)	3.8 (2.9–7.5)	0.004

N(%), median (IQR), Mann–Whitney U test, chi-square test, or Fisher’s exact test. AFP, alpha-fetoprotein; HCV, hepatitis C virus; MELD, model for end-stage liver disease; APRI, AST to platelet ratio index; FIB-4, fibrosis-4 index.

**Table 2 medicina-60-01132-t002:** Changes in liver function.

	Child–Pugh	*p*-Value (vs. Baseline)	MELD	*p*-Value (vs. Baseline)
Baseline	5.0 (5.0–5.0)		7.0 (6.0–8.0)	
1 year	5.0 (5.0–5.0)	0.498	7.0 (6.0–8.0)	0.009
2 year	5.0 (5.0–5.0)	0.094	7.0 (6.0–7.0)	<0.001
3 year	5.0 (5.0–5.0)	0.292	7.0 (6.0–7.0)	<0.001
4 year	5.0 (5.0–5.0)	0.134	6.0 (6.0–7.0)	<0.001
5 year	5.0 (5.0–5.0)	0.315	6.0 (6.0–7.0)	0.001
6 year	5.0 (5.0–5.0)	0.317	7.0 (6.0–7.0)	0.005
7 year	5.0 (5.0–5.0)	0.414	7.0 (7.0–8.0)	0.109

Median (IQR), Wilcoxon signed rank test. MELD, model for end-stage liver disease.

**Table 3 medicina-60-01132-t003:** Changes in noninvasive fibrosis score.

	APRI	*p*-Value (vs. Baseline)	FIB-4	*p*-Value (vs. Baseline)
Baseline	0.5 (0.3–1.1)		2.6 (1.7–4.9)	
1 year	0.4 (0.3–0.6)	<0.001	2.1 (1.5–3.2)	<0.001
2 year	0.4 (0.3–0.5)	<0.001	2.1 (1.5–3.1)	<0.001
3 year	0.4 (0.3–0.6)	<0.001	2.1 (1.5–3.1)	<0.001
4 year	0.4 (0.3–0.5)	<0.001	2.1 (1.5–3.3)	<0.001
5 year	0.4 (0.3–0.6)	<0.001	2.1 (1.5–3.2)	<0.001
6 year	0.6 (0.3–0.6)	<0.001	2.2 (1.6–3.6)	<0.001
7 year	0.5 (0.3–0.8)	<0.001	2.9 (2.0–5.0)	0.006

Median (IQR), Wilcoxon signed rank test. APRI, AST to platelet ratio index; FIB-4, Fibrosis-4 index.

**Table 4 medicina-60-01132-t004:** Characteristics of HCC patients.

Patient	Gender	Age	Genotype	Treatment	Cirrhosis	Alcohol	Time after SVR (Months)	AFP (ng/mL)	Child–Pugh	MELD	APRI	FIB-4
1	Male	73	1b	LDV/SOF	+	–	25	6.0	5	7	0.490	2.751
2	Female	48	2a	SOF + RBV	+	–	18	4.31	6	12	3.871	13.569
3	Male	41	3a	SOF + RBV	+	+	75	7.18	5	6	0.878	1.651
4	Male	76	2a	SOF + RBV	+	–	3	9.7	5	6	0.452	2.867
5	Male	69	1b	SOF + DCV	+	–	27	8.02	5	8	0.930	3.825
6	Female	56	2a	SOF + RBV	+	+	51	5.61	5	6	0.833	3.528
7	Male	67	2a	SOF + RBV	+	–	17	4.29	5	9	0.951	2.833
8	Female	71	2a	SOF + RBV	–	–	37	3.8	5	6	0.693	3.377
9	Male	69	3a	SOF + DCV	+	–	68	5.93	5	6	0.675	3.290
10	Male	67	2a	SOF + RBV	–	–	28	8.35	5	6	0.717	3.630
11	Female	70	1b	LDV/SOF	+	–	73	2.7	5	7	0.689	4.112
12	Female	63	2a	SOF + RBV	+	–	50	6.35	5	8	3.202	12.028
13	Female	70	2	SOF + RBV	+	–	86		5	9	2.627	6.859
14	Female	71	2a/2c	SOF + RBV	+	+	6		5	9	0.714	5.626
15	Male	79	2	SOF + RBV	+	–	21	0.97	5	7	0.747	7.469
16	Male	70	2	SOF + RBV	–	–	46	2.01	5	8		
17	Male	78	2a/2c	SOF + RBV	–	–	2	4.8	5	8	0.694	5.594
18	Male	54	2	SOF + RBV	+	–	65	9.99	5	7	2.786	11.580
19	Male	80	2	SOF + RBV	+	–	49	13.18	5	7	1.472	11.424
20	Female	66	2	SOF + RBV	+	–	30	178.46	5	6	5.69	10.951
21	Female	69	1b	LDV/SOF	+	–	26	45.08	5	7	0.974	2.465

SVR, sustained virologic response; AFP, alpha-fetoprotein; MELD, model for end-stage liver disease; APRI, AST to platelet ratio index; FIB-4, fibrosis-4 index; LDV, ledipasvir; SOF, sofosbuvir; RBV, ribavirin; DCV, daclatasvir.

**Table 5 medicina-60-01132-t005:** Univariate and multivariate analysis of the risk factors for the development of HCC following SVR.

Variable	Univariate Analysis		Multivariate Analysis (Age, Cirrhosis, AFP)		Multivariate Analysis (Age, AFP, APRI)	
	HR (95% CI)	*p*-Value	HR (95% CI)	*p*-value	HR (95% CI)	*p*-Value
Age > 65 years	4.761 (1.730–13.102)	0.003	3.547 (1.263–9.962)	0.016	3.182 (1.120–9.040)	0.030
Cirrhosis	6.912 (2.300–20.768)	0.001	4.916 (1.607–15.042)	0.005		
AFP > 4.2 ng/mL	5.452 (1.809–16.433)	0.003			3.789 (1.027–13.984)	0.046
APRI > 0.670	11.739 (2.710–50.843)	0.001			4.896 (1.026–23.366)	0.046

HCC, hepatocellular carcinoma; CI, confidence interval; HR, hazard ratio; SVR, sustained virological response.

## Data Availability

Data are contained within the article and Appendix A.

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
