# Peer review of "Prognosis Following Sustained Virologic Response in Korean Chronic Hepatitis C Patients Treated with Sofosbuvir-Based Treatment: Data from a Multicenter Prospective Observational Study up to 7 Years"

_medicina, 2024, doi:10.3390/medicina60071132_

Round 1
Reviewer 1 Report
Comments and Suggestions for Authors
First of all, I would like to congratulate the authors for their work. They have a good cohort of patients and more than enough follow-up time.
I have a doubt that I think they should clarify in the methodology: why do they use medians in the continuous variables? They do not provide information on having carried out a previous study of the normality of the variables. It would be interesting if they could justify this. Moreover, they give a result of a median age, when according to what they say they use medians and interquartile ranges.
Author Response
First of all, I would like to congratulate the authors for their work. They have a good cohort of patients and more than enough follow-up time.
I have a doubt that I think they should clarify in the methodology: why do they use medians in the continuous variables? They do not provide information on having carried out a previous study of the normality of the variables. It would be interesting if they could justify this. Moreover, they give a result of a median age, when according to what they say they use medians and interquartile ranges.
Response: Thank you very much for your kind review. We added the introduction of ‘statistical analaysis’ and correct age in result as follows
We used descriptive statistics to describe the baseline demographics of the patients. Because data are not normally distributed, continuous variables are expressed as medians (interquartile ranges), while categorical variables are presented as numbers (%).
We analyzed 516 Korean patients with CHC who achieved sustained SVR after SOF or ledipasvir/SOF treatment. The cohort included 44.8% males, with a median age of 62 years. (Table 1)
Reviewer 2 Report
Comments and Suggestions for Authors
This manuscript describes the prognosis of sustained virological response in Korean patients with chronic hepatitis C receiving sofosbuvir-based therapy. A total of 516 patients were included in this analysis under a 7-year dataset from a multicenter prospective observational study. Cirrhosis was observed in 160 patients; HCC occurred in 21 patients. Patients with HCC were older and had a higher cirrhosis prevalence; higher AFP and higher APRI. Cox regression analysis showed that age > 65 years and cirrhosis were significant risk factors for HCC. Patients who achieve SVR after sorafenib treatment generally have a good prognosis. However, the risk has not been eliminated, especially in the elderly and patients with cirrhosis. Therefore, regular follow-up, monitoring, and early treatment are needed. I really admire the author's long-term and continuous research. The manuscript is also well writing and represents an interesting work. As such,
1. Correction sequence of “2.2 Patients” and “2.1 Statistical analysis”;
2. About “2.1 Statistical analysis”, Can the author give more detailed introduction;
3. Regarding the description of the trend in the article, e.g. “Liver function and fibrosis markers showed notable trends over time”., the author can add some curve graphs to present it more intuitively;
4. Regarding the genotype 1/2/3, the author need to give more information.
5. References need to be in a standardized format. E.g. References 6, “e3” unnecessary; page, reference 8 and 12, page format different.
Author Response
This manuscript describes the prognosis of sustained virological response in Korean patients with chronic hepatitis C receiving sofosbuvir-based therapy. A total of 516 patients were included in this analysis under a 7-year dataset from a multicenter prospective observational study. Cirrhosis was observed in 160 patients; HCC occurred in 21 patients. Patients with HCC were older and had a higher cirrhosis prevalence; higher AFP and higher APRI. Cox regression analysis showed that age > 65 years and cirrhosis were significant risk factors for HCC. Patients who achieve SVR after sorafenib treatment generally have a good prognosis. However, the risk has not been eliminated, especially in the elderly and patients with cirrhosis. Therefore, regular follow-up, monitoring, and early treatment are needed. I really admire the author's long-term and continuous research. The manuscript is also well writing and represents an interesting work. As such,
Comment 1. Correction sequence of “2.2 Patients” and “2.1 Statistical analysis”;
Response 1: Thanks for your kind review. We corrected this error.
Comment 2. About “2.1 Statistical analysis”, Can the author give more detailed introduction;
Response 2: Thank you very much for your kind review. We added the introduction of ‘statistical analaysis’ as follows
We used descriptive statistics to describe the baseline demographics of the patients. Because data are not normally distributed, continuous variables are expressed as medians (interquartile ranges), while categorical variables are presented as numbers (%).
Comment 3. Regarding the description of the trend in the article, e.g. “Liver function and fibrosis markers showed notable trends over time”., the author can add some curve graphs to present it more intuitively;
Response 3: Thank you very much for your suggestion. Even if the changes in the variables you suggested were statistically significant, it would be difficult to graph the numerical differences, so we have omitted them.
Comment 4. Regarding the genotype 1/2/3, the author need to give more information.
Response4 : Thank you for your suggestion. We tried to show subtype for genotype, but in some cases there was no subtype result, so we omitted it.
Comment 5. References need to be in a standardized format. E.g. References 6, “e3” unnecessary; page, reference 8 and 12, page format different.
Response 5: Thanks for your kind review. We corrected references.